# The association between the triglyceride-glucose index with all-cause and cardiovascular mortality within the infertility population

**Yuhan Wang[1], Yishu Tian[2], Feifei Zhou** **[1]\*, Zixing Zhong[3]\***

**1** Department of Reproductive Endocrinology, Center for Reproductive Medicine, Zhejiang Provincial People's Hospital (Affiliated People's Hospital, Hangzhou Medical College), Hangzhou, Zhejiang, China, **2** Department of Ultrasound Medicine, Center for Reproductive Medicine, Zhejiang Provincial People's Hospital (Affiliated People's Hospital, Hangzhou Medical College), Hangzhou, Zhejiang, China, **3** Department of Obstetrics, Center for Reproductive Medicine, Zhejiang Provincial People's Hospital (Affiliated People's Hospital, Hangzhou Medical College), Hangzhou, Zhejiang, China

\* 5510025@zju.edu.cn (FZ); zhongzixing@hmc.edu.cn (ZZ)

## Abstract

### Background

The relationship between triglyceride-glucose (TyG) index and all-cause or cardiovascular mortality among infertile women remains unclear. In this study, we intended to utilize a national cohort from National Health and Nutrition Examination Survey (NHANES) to check the association between them.

### Methods

Ten datasets from the NHANES database spanning almost 20 years were used as the data source and were combined within National Death Index for mortality follow-up. Multiple-variable Cox proportionate hazards regression models and three others were employed in this study to for assessing relationships among TyG index levels with all-cause and cardiovascular mortality. SPSS (version 29.0) and online websites were utilized for conducting the primary statistical analyses.

### Results

1,450 female participants were identified in this study. The samples were classified based on TyG index quartiles (7.05–11.95). The TyG index had a mean of 8.58±0.66. Participants with higher TyG indices were older-aged, had greater body mass index (BMI), and a stronger likelihood of having hypertension and diabetes ($P < 0.05$). Participants whose TyG indices were higher were older in age, along with increased BMI, and blood pressure along with diabetes ($P < 0.05$). Significant positive associations were observed among the TyG index and total mortality in the crude model (HR: 1.81, 95% CI: 1.27–2.58). Correlation persisted in Model 2 (following the adjustment

**Data availability statement:** The datasets generated and/or analyzed during the current study are available in the National Health and Nutrition Examination Survey (NHANES) repository: https://www.cdc.gov/nchs/nhanes/index.htm.

**Funding:** This study was supported by the Zhejiang Provincial Project for Medical and Health Science (No. 2025KY027) which was received by Zixing Zhong. The funder had no role in study design, data collection and analysis, decision to publish, or preparation of the manuscript.

**Competing interests:** The authors declare no competing interests.

of age and race) and Model 3 (following the adjustment of age, race, BMI, education, family poverty income ratio, smoking and drinking habits, menstrual regularity, hypertension, and diabetes). The TyG index did not affect the cardiovascular mortality in infertile women.

## Conclusion

TyG index levels were in positive association with all-cause mortality within the female infertile population.

## Introduction

Infertility is defined as being unable to conceive after ≥ 12 months of regular unprotected sexual intercourse or because of a reduced capacity to reproduce either as an individual or with a partner [1]. Infertility affects up to 9% of women of reproductive age worldwide [2]. Infertility is a complex, multifaceted issue, which may have profound long-term impacts on women's physical and emotional health. Current research has identified that infertility is associated with several long-term complications, including metabolic disorders, psychological stress, and an increased risk of certain cancers [3,4]. Emerging evidence also suggests that infertility may be linked to long-term cardiovascular risks [5,6]. Women with infertility often have underlying conditions such as polycystic ovary syndrome (PCOS), which is associated with insulin resistance (IR), dyslipidemia, and chronic inflammation-key contributors to cardiovascular disease (CVD) [7]. Additionally, infertility-related psychological stress and the use of assisted reproductive technologies may further exacerbate CVD risk through mechanisms such as endothelial dysfunction and thromboembolic events [6]. Nulliparity, common among infertile women, has also been associated with higher CVD risk compared to parous women [8].

CVDs rank as the top cause of disease-related death worldwide, with increasing incidence in recent decades [9,10]. The latest Global Burden of Disease (GBD 2019) Study revealed that the prevalence of CVDs has almost doubled, with an increase to 523 million cases in 2019 from 271 million cases in 1990 [10]. Therefore, it is crucial to identify indicators that identify individuals with increased cardiovascular risks. A few studies have suggested an association of infertility with cardiovascular risks and mortality [11–13].

Triglyceride–glucose (TyG) index is a pragmatic method to assess metabolic health by measuring fasting blood triglyceride and glucose levels, and a critical calculable biomarker to identify IR, that defined indicator of CVD. The TyG index is effective for evaluating CVD risks and predicting cardiovascular mortality [14]. Some studies have suggested that an elevated TyG index has an independent correlation to adverse prognosis for acute cardiovascular events [15,16].

However, there is limited research on how TyG index relates to all-cause deaths in infertility individuals. Therefore, this present research was designed to explore the relationship that exists among TyG index levels and all-cause or cardiovascular

mortality in infertile women. As far as we know, this is the first study to utilize National Health and Nutrition Examination Survey (NHANES) database for investigating whether or not TyG index is capable of predicting all-cause or cardiovascular mortality for infertile populations.

## Materials and methods

### Data source

This study utilized the NHANES 1999–2018. The NCHS authorized human subjects to be studied, with each participant approving the informed consent prior to examination and survey. The data cover ten cycles: 1999–2000, 2001–2002, 2003–2004, 2005–2006, 2007–2008, 2009–2010, 2011–2012, 2013–2014, 2015–2016, and 2017–2018.

### Study population

The data span a period of nearly two decades. The inclusion criteria were women with fasting blood glucose and triglyceride data and who answered the Reproductive Health Questionnaire (RHQ). Initially, this study enrolled all eligible female participants (n = 13,000). The exclusion criteria were participants without reproductive data from the RHQ, women with missing covariants, and those without follow-up data for mortality status. Finally, 1450 participants were included (Fig 1).

### TyG index evaluation

By the way, TyG was the major study variant. The value was calculated according to the formula: Ln[triglycerides (mg/dl)*-fasting glucose (mg/dl)/2]. Triglyceride with glucose levels measured using an automatic biochemistry analyzer. Participants were required to fast for eight hours before the measurement.

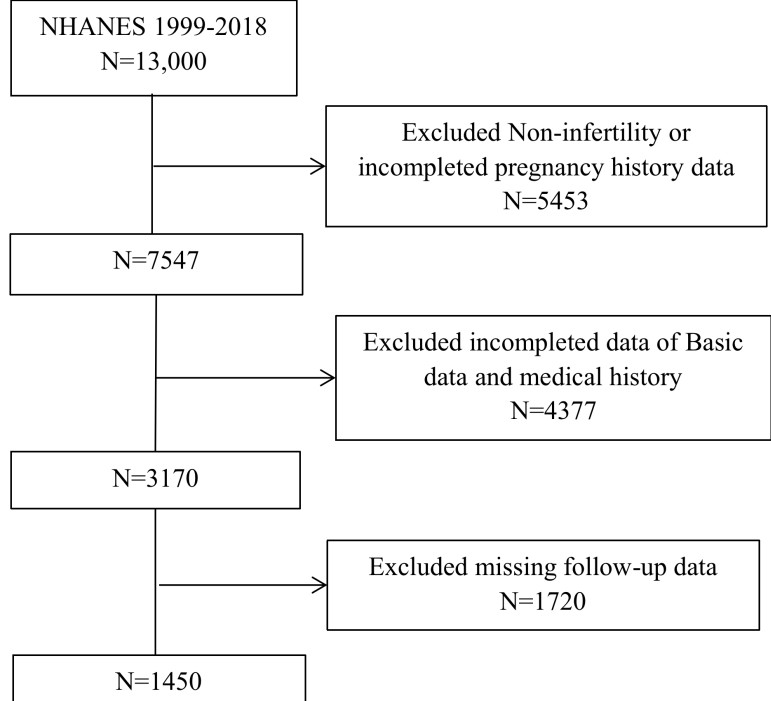

**Fig 1. Flowchart of NHANES (1999-2018) sample selection.** After screening, a total number of 1450 participants were enrolled. NHANES: National Health and Nutrition Examination Survey.

## Assessment of infertility

For the cycles before 2013, infertility was defined as women who were not single and had never been pregnant or had a live birth. For the 2013–2018 cycles, infertility was defined by means of the following question: "Have you ever tried to conceive for at least a year without becoming pregnant?" (RHQ074). Those who gave the answer "yes" were seen as infertile and enrolled in this cohort.

## Assessment of covariates

This study also included other covariates to comprehensively describe relationship between TyG index and infertile women's mortality. Variables such as age, race, educational status, body mass index (BMI), marriage condition, family poverty income ratio (PIR), drinking and smoking situation were considered. Data on medical history, including hypertension, diabetes, CVDs (heart failure, angina, heart attack, coronary disease), pulmonary diseases (PDs) (emphysema, asthma, chronic bronchitis), liver condition, and cancer status, were extracted.

## Assessment of mortality

The research used the NDI, NHANES public use-connected mortality data, updated in 2019. The identification of disease-specific deaths was carried out by employing the ICD-10. CVD deaths encompass those resulting from heart diseases (ICD-10 codes I00-I09, I11, I13, I20-I51), primary blood pressure and hypertonic kidney disease (I10, I12, I15), as well as brain vascular illnesses (I60-I69).

## Ethics approval and consent to participate

The studies involving human participants were reviewed and approved by the NCHS Research Ethics Review Board (ERB).

## Statistical analysis

The statistics here were performed based on the guidelines of the CDC. Author adopted SPSS (version 29.0) and an online database (www.mimicdb.com) to facilitate variable extraction and statistical analyses. Categorical variables are shown as percentages and compared using the chi-square ($\chi^2$) test or Fisher's exact test when expected frequencies were low. Continuous variables were assessed for normality using the Shapiro-Wilk test, with normally distributed variables expressed as mean ± standard deviation (SD) and compared using Student's t-test or one-way ANOVA for multiple groups, while non-normally distributed variables were expressed as median (interquartile range, IQR) and compared using the Mann-Whitney U test or Kruskal-Wallis test for multiple groups. Some continuous variables were transformed into ordinal variables; for example, the BMI was divided into "1" for participants whose BMI ≥ 30 kg/m² and "2" for those considered within the normal range. Regarding marital status, both "married (code 1)" and "living with partner (code 6)" were considered "married", whereas the others were considered "single". To evaluate the association between the TyG index and all-cause and CVD mortality, multivariable Cox proportional hazards regression models were employed. Three models were constructed: Model 1 (Crude Model) was unadjusted; Model 2 adjusted for age and race; and Model 3 further adjusted for BMI, education level, PIR, smoking and drinking habits, menstrual regularity, hypertension, and diabetes. Hazard ratios (HRs) and 95% confidence intervals (CIs) were calculated to quantify the strength of the associations, and the proportional hazards assumption was tested using Schoenfeld residuals, with no violations detected. Participants were stratified into quartiles based on their TyG index values (Q1: 7.05–8.12; Q2: 8.12–8.51; Q3: 8.51–8.99; Q4: 8.99–11.95), and the association between TyG index quartiles and mortality outcomes was assessed using Cox regression models.

The lowest quartile (Q1) was regarded as the reference group. Sensitivity analyses were performed to assess the robustness of the findings, including excluding participants with pre-existing CVD or cancer and conducting stratified

analyses by age groups (<45 years vs. ≥ 45 years) and BMI categories (obese vs. non-obese). Missing data for covariates were handled using multiple imputation with chained equations (MICE) to create five imputed datasets, and the results were pooled using Rubin's rules. A two-tailed p-value < 0.05 was considered statistically significant for all analyses, and all statistical tests were validated using the online database (www.mimicdb.com) to ensure reproducibility.

## Results and discussion

### Basic features of participants

Finally, 1,450 participants are included after excluding more than 10,000 participants due to missing variables. Mean TyG index was 8.58±0.66, with TyG index quartiles at 7.05–8.12, 8.12–8.51, 8.51–8.99, and 8.99–11.95. The mean age of the entire set of participants was 45.40±18.59 years, with an average BMI of 29.07±7.60 kg/m$^2$. Moreover, 82.55% of participants had an educational level of high school or above, 80.65% had good family income (PIR ≥ 1.1), 30.21% had hypertension, 8.69% had diabetes, and 8.21% had cancer. Furthermore, 64.34% of the infertile participants drank alcohol, while 32.5% were current smokers. Significant differences were observed in age, BMI, race, levels of education, marital status, alcohol habit, diabetes, hypertension, CVD, and cancer status ($P < 0.05$). Individuals with larger TyG scores had a much greater likelihood of being Mexican-American. or white, have concurrent hypertension and diabetes, and be older and have higher BMIs than those who had a 1st quartile TyG index. The numbers of live births, stroke risk, PDs, and liver diseases did not vary across the quartiles ($P > 0.05$). Table 1 shows the findings in detail.

### The association of TyG index level for all-cause or CVD mortality

The mean follow-up period was 142.80 months. Regarding mortality, there were 221 all-cause mortalities along with 77 cardiovascular mortalities. Three models have been established to investigate the independent associations among TyG index and all-cause and cardiovascular mortality for infertile individuals Detailed data are presented in Table 2. In Model 1, TyG index was significantly and positively associated with risk of all-cause death (HR: 1.81, 95% CI: 1.27–2.58). After adjusting participants' age and race in Model 2, TyG index continued to be directly related to all-cause mortality (HR: 1.83, 95% CI: 1.27–2.58). Similarly, TyG index had a positive interaction with all-cause mortality hazard following adjustment for age, race, BMI, education level, PIR, smoking and drinking habits, menstrual regularity, hypertension, and diabetes (HR: 1.69, 95% CI: 1.05–2.72). However, when participants were sub-grouped relying on TyG Index quartiles, only Model 1 showed a positive association. In comparison with individuals in its lowest quartile, those in quartile 3 presented greater all-cause hazard of death (HR: 2.11, 95% CI: 1.17–3.80). No associations were observed in Models 2 and 3. No association was found between TyG index in relation to CVD death. However, in Model 1, when divided into quartiles, HRs (95% CIs) for cardiovascular mortality in quartiles 2, 3, and 4 for TyG index were 3.99 (1.29–12.32), 4.46 (1.32–15.01), and 3.78 (0.82–17.49), respectively. The quartile-based link for TyG index and CVD death persists, yet no statistical significance was detected in Model 2. It was found that no relation existed in adjusted Model 3 between them.

This study included 1450 female infertile participants, the mortality status of whom was followed up for more than a decade on average. Numerous articles investigated relationship between TyG index and CVD. Da Silva et al. demonstrated evidence of a normal correlation of TyG index and symptomatic coronary heart disease (CHD), indicating its potential role as a marker for atherosclerosis [17]. The outcomes for the MIMIC-III databank indicated that TyG index represents an effective predictor of higher death among critically ill individuals with CHD [18]. For patients with non-alcoholic fatty liver disease, a rise in TyG index is linked to potential CHD risk, reflecting the severity of coronary atherosclerosis [19]. Wang et al. confirmed TyG index was an independent contributor for multivessel coronary artery disease (CAD) after adjusting confounders [20]. Chen et al., also proposed TyG index could forecast CAD severity [20,21]. A longitudinal double-cohort study identified a remarkable link at baseline between TyG index (β for c-f PWV = 0.61, $p = 0.018$) and increased arterial stiffness, as well as a positive correlation with stroke incidence, overall-cause fatality, and cardiovascular mortality rates [22]. Zhao et al. stated abnormally raised TyG index levels were related to chest pain onset, and its

**Table 1. Weighted baseline characteristics of the study population.**

| | All participants | Quartile 1 | Quartile 2 | Quartile 3 | Quartile 4 | P-value |
|---|---|---|---|---|---|---|
| | N = 1450 | N = 360 | N = 359 | N = 365 | N = 366 | |
| | | 7.05–8.12 | 8.12–8.51 | 8.51–8.99 | 8.99–11.95 | |
| Age (year) | 45.40 (18.59) | 39.44 (14.96) | 45.08 (18.74) | 48.18 (20.26) | 48.79 (18.54) | < 0.001 |
| BMI (kg/m²) | 29.07 (7.60) | 26.29 (6.88) | 28.44 (7.70) | 29.62 (7.47) | 31.88 (8.28) | < 0.001 |
| BMI, % | | | | | | < 0.001 |
| Follow-up time (months) | 142.80 (67.59) | 144.69 (72.20) | 142.09 (67.40) | 140.20 (66.61) | 144.23 (64.16) | 0.582 |
| Obesity (≥ 30 kg/m²) | 37.10 | 20.83 | 36.49 | 38.63 | 52.19 | |
| Normal | 62.90 | 79.17 | 63.51 | 61.37 | 47.81 | |
| livebirth (times) | 0.73 (0.81) | 0.80 (1.01) | 0.70 (0.75) | 0.71 (0.76) | 0.70 (0.70) | 0.261 |
| Races, % | | | | | | < 0.001 |
| Mexican American | 14.62 | 10.56 | 11.70 | 17.53 | 18.58 | |
| Other Hispanic | 7.24 | 6.39 | 8.91 | 7.40 | 6.28 | |
| Non-Hispanic white | 51.17 | 46.67 | 49.58 | 52.05 | 56.28 | |
| Non-Hispanic Black | 20.28 | 29.17 | 23.68 | 15.07 | 13.39 | |
| Others | 6.69 | 7.22 | 6.13 | 7.95 | 5.46 | |
| Educational levels, % | | | | | | < 0.001 |
| Less than 9th grade | 6.69 | 2.22 | 5.57 | 7.67 | 11.20 | |
| 9–11th grade | 10.76 | 7.50 | 11.70 | 12.05 | 11.75 | |
| High school graduate | 21.17 | 16.11 | 22.28 | 21.64 | 24.59 | |
| Some college or AA degree | 32.97 | 34.17 | 32.31 | 32.88 | 32.51 | |
| College graduate or above | 28.41 | 40.00 | 28.13 | 25.75 | 19.95 | |
| Marrige, % | | | | | | < 0.001 |
| Married | 75.38 | 67.78 | 75.49 | 77.81 | 80.33 | |
| Single | 24.62 | 32.22 | 24.51 | 22.19 | 19.67 | |
| PIR, % | | | | | | 0.018 |
| Poor household income (≤ 1.1) | 19.35 | 15.04 | 19.89 | 18.23 | 24.18 | |
| High household income (> 1.1) | 80.65 | 84.96 | 80.11 | 81.77 | 75.82 | |
| Smoking, % | 32.50 | 37.60 | 38.08 | 45.36 | 38.41 | 0.01 |
| Alcohol use, % | 64.34 | 71.11 | 64.90 | 60.82 | 60.66 | 0.01 |
| Hypertension, % | 30.21 | 17.78 | 28.97 | 35.89 | 37.98 | < 0.001 |
| Diabetes, % | 8.69 | 2.22 | 3.90 | 6.85 | 21.58 | < 0.001 |
| CVD, % | 4.00 | 1.67 | 2.79 | 6.03 | 5.46 | 0.006 |
| Stroke, % | 2.34 | 0.83 | 1.95 | 3.56 | 3.01 | 0.075 |
| PD, % | 20.62 | 18.06 | 22.01 | 19.18 | 23.22 | 0.278 |
| Liver condition, % | 2.55 | 2.22 | 2.79 | 1.64 | 3.55 | 0.404 |
| Cancer, % | 8.21 | 6.11 | 5.85 | 9.59 | 11.20 | 0.018 |
| MC, % | | | | | | < 0.001 |
| Regular | 50.07 | 65.00 | 52.09 | 45.75 | 37.70 | |
| Irregular | 49.93 | 35.00 | 47.91 | 54.25 | 62.30 | |

BMI, Body mass index; PIR, poverty income ratio; CVD, Cardiovascular disease; PD, Pulmonary disease; MC, Menstrual cycle

correlation with all-cause mortality regardless of whether chest pain was present [23]. An examination of 12 cohort studies, conducted through a meta-analysis approach, suggested a possible link between elevated TyG index values and a heightened risk of CVD within the general populace [24]. A forward-looking investigation carried out in China pointed out

**Table 2. HRs (95% CI) for mortality according to the TyG index.**

| | HR (95% CI) *P* value | | |
| | Model 1 | Model 2 | Model 3 |
|---|---|---|---|
| **All-cause mortality** | | | |
| TyG index (continuous) | 1.81 (1.27, 2.58) 0.001 | 1.83 (1.17, 2.87) 0.009 | 1.69 (1.05, 2.72) 0.031 |
| TyG index (quartiles) | | | |
| Quartile 1 | Reference | Reference | Reference |
| Quartile 2 | 1.77 (1.02, 3.07) 0.04 | 0.91 (0.52, 1.62) 0.75 | 0.93 (0.52, 1.66) 0.81 |
| Quartile 3 | 2.11 (1.17, 3.80) 0.01 | 0.73 (0.39, 1.40) 0.34 | 0.71 (0.37, 1.37) 0.31 |
| Quartile 4 | 1.59 (0.73, 3.45) 0.26 | 0.63 (0.25, 1.55) 0.63 | 0.55 (0.22, 1.38) 0.20 |
| **Cardiovascular mortality** | | | |
| TyG index (continuous) | 1.48 (0.75, 2.89) 0.26 | 1.34 (0.59, 3.08) 0.486 | 0.95 (0.38, 2.40) 0.92 |
| TyG index (quartiles) | | | |
| Quartile 1 | Reference | Reference | Reference |
| Quartile 2 | 3.99 (1.29, 12.32) 0.02 | 2.26 (0.71, 7.19) 0.17 | 2.45 (0.76, 7.93) 0.14 |
| Quartile 3 | 4.46 (1.32, 15.01) 0.02 | 1.77 (0.48, 6.46) 0.39 | 1.98 (0.52, 7.55) 0.32 |
| Quartile 4 | 3.78 (0.82, 17.49) 0.09 | 1.90 (0.34, 10.66) 0.47 | 2.19 (0.36, 13.57) 0.40 |

Model 1: no covariates were adjusted; Model 2: only age and race were adjusted; Model 3: age, race, BMI, education, PIR, smoking and drinking habits, menstrual regularity, hypertension, and diabetes were adjusted.

BMI, Body mass index; PIR, poverty income ratio.

a relation between TyG index and the stroke risks in middle-aged and elderly people, accounting for over half of the total association between BMI and stroke in this age bracket [25]. For critical ischemic stroke patients, evidence suggests a gradual rise in all-cause mortality along with a increasing TyG index [16].

However, some studies have suggested conflicting results. Alizargar et al. concluded that the role of TyG in CVD is inconsistent across different sub-types [25]. A prospective study involving populations from five continents demonstrated no apparent link among TyG index and non-CVD death rates, with no link to cardiovascular mortality in high-income countries [26]. Moreover, some studies have suggested this U-shaped connection of TyG index and all-cause mortality in CVD people [27]. Different conclusions may arise from factors such as age, heterogeneous patient populations, geographical location, and economic disparities [28].

The mechanisms by which TyG leads to increased mortality rates remain unclear [28]. IR may be the key mediator bridging the TyG index with CVD advancement [29]. Insulin is important to increase blood flow by mediating the production of nitric oxide (NO) in the vascular endothelium, thereby enhancing glucose processing. Reduced NO production in patients with IR is vital for the development of CVD. In addition, IR leads to increased vascular stiffness and fibrosis, promoting CVD; however, the intricate mechanisms and mediators involved are complex. Studies have also indicated an association between IR and hypertension, with atherosclerosis as a potential consequence [30]. In addition, IR is a common feature in PCOS, which is a significant contributor to female infertility. PCOS is considered to have higher CVD risks and mortality in the long term [31].

## Strengths and limitations

Overall, several benefits are present in this study. First, the data were extracted from a national database, including 1450 infertile women, representing a large sample size. Second, this research focused on lipid imbalance and its role in mortality in infertile women, which has not been previously studied. This study evaluated the connection between TyG index and all-cause and cardiovascular mortality in female infertile individuals. However, there are some limitations. First, the nature of the study design precludes further exploration of the causal effects of higher TyG levels on the future health outcomes

of infertile women. Second, infertility was determined by self-report interviews or their obstetric history; this may be less accurate than a laboratory diagnosis, which has the potential to compromise the conclusion. Nevertheless, this study assessed the indicator with several covariates to ensure that the data were not outliers. Third, although the study adjusted for a range of covariates, it was unable to entirely eliminate the impact of unnoticed confounders. Further clinical research with larger cohorts is necessary to confirm the link between the TyG index and mortality in infertile women.

## Conclusion

In conclusion, higher TyG index levels are linked to greater all-cause mortality. This correlation remained notable even after accounting for multiple covariates, indicating TyG index might be an independent indicator for predicting all-cause deaths. However, the TyG index appears to be unrelated to cardiovascular mortality, and its ability to predict cardiovascular mortality is debatable. These findings offer a unique perspective on long-term healthcare and highlight infertile populations.

## Acknowledgments

The authors thank the staff and the participants of the NHANES study for their valuable contributions.

## Author contributions

**Data curation:** Yuhan Wang, Yishu Tian.

**Writing – original draft:** Yuhan Wang, Yishu Tian.

**Writing – review & editing:** Feifei Zhou, Zixing Zhong.

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
