## [Editor Report · Decision Letter 0]

7 Jan 2025

PONE-D-24-58493The association between the triglyceride-glucose index with all-cause and cardiovascular mortality within the infertility populationPLOS ONE

Dear Dr. Zhou,

Thank you for submitting your manuscript to PLOS ONE. After careful consideration, we feel that it has merit but does not fully meet PLOS ONE’s publication criteria as it currently stands. Therefore, we invite you to submit a revised version of the manuscript that addresses the points raised during the review process.

We look forward to receiving your revised manuscript.

Kind regards,

Marwan Al-Nimer

Academic Editor

PLOS ONE

**Journal Requirements:**

This study was co-supported by the Zhejiang Provincial Project for Medical and Health Science (No. 2022503241) and Zhejiang Provincial Project for Education (No. Y202249319).

**Additional Editor Comments:**

This interesting study highlights the relationship between the TyGI and the all mortality in the infertile women. there are few points need to be clarify

1. Introduction: A paragraph explaining why infertile women are at risk for cardiovascular events. Add the rationale of the study

2. Methods: why the authors collected the data up to 2018. The statistic analysis requires more details as the results showed findings with specific statistical tests.

3. Results: The flowchart (figure 1) is without legend and the tables without footnotes explaining the table thoroughly.

4. References: many typing errors

5. The overall abbreviations require major revision

---

## [Author Response · Author response to Decision Letter 1]

17 Feb 2025

Dear Marwan Al-Nimer,

Thank you for your valuable feedback and constructive suggestions regarding our manuscript. We have carefully reviewed the comments and have made the necessary revisions to address the points raised.

---

## [Editor Report · Decision Letter 1]

20 Feb 2025

The association between the triglyceride-glucose index with all-cause and cardiovascular mortality within the infertility population

PONE-D-24-58493R1

Dear Dr. Feifei Zhou

We’re pleased to inform you that your manuscript has been judged scientifically suitable for publication and will be formally accepted for publication once it meets all outstanding technical requirements.

Kind regards,

Marwan Al-Nimer

Academic Editor

PLOS ONE
---

## [Editor Report · Acceptance letter]

PONE-D-24-58493R1

PLOS ONE

Dear Dr. Zhou,

I'm pleased to inform you that your manuscript has been deemed suitable for publication in PLOS ONE. Congratulations! Your manuscript is now being handed over to our production team.

Kind regards,

on behalf of

Dr. Marwan Al-Nimer

Academic Editor

PLOS ONE